# Optimal Proactive Caching for Multi-View Streaming Mobile Augmented Reality

**Zhaohui Huang * and Vasilis Friderikos**

Department of Engineering, King's College London, London WC2R 2LS, UK; vasilis.friderikos@kcl.ac.uk
* Correspondence: zhaohui.huang@kcl.ac.uk

**Abstract:** Mobile Augmented Reality (MAR) applications demand significant communication, computing and caching resources to support an efficient amalgamation of augmented reality objects (AROs) with the physical world in multiple video view streams. In this paper, the MAR service is decomposed and anchored at different edge cloud locations to optimally explore the scarce edge cloud resources, especially during congestion episodes. In that way, the proposed scheme enables an efficient processing of popular view streams embedded with AROs. More specifically, in this paper, we explicitly utilize the notion of content popularity not only to synthetic objects but also to the video view streams. In this case, popular view streams are cached in a proactive manner, together with preferred/popular AROs, in selected edge caching locations to improve the overall user experience during different mobility events. To achieve that, a joint optimization problem considering mobility, service decomposition, and the balance between service delay and the preference of view streams and embedded AROs is proposed. To tackle the curse of dimensionality of the optimization problem, a nominal long short-term memory (LSTM) neural network is proposed, which is trained offline with optimal solutions and provides high-quality real-time decision making within a gap between 5.6% and 9.8% during inference. Evidence from a wide set of numerical investigations shows that the proposed set of schemes owns around 15% to 38% gains in delay and hence substantially outperforms nominal schemes, which are oblivious to user mobility and the inherent multi-modality and potential decomposition of the MAR services.

**Keywords:** 5G; augmented reality; view streams; mobility; edge cloud; long short-term memory neural network

## 1. Introduction

Mobile Augmented Reality (MAR) has recently attracted significant attention from both industry and academia due to the fact that it can enhance physical world experiences by amalgamating digital Augmented Reality Objects (AROs). Despite the unlimited capabilities in building hybrid worlds, MAR applications are demanding in computing and caching resources especially when rendering large and complex three-dimensional (3D) AROs [1]. Edge clouds (ECs) can satisfy the requirements of MAR applications because they enable the deployment of computing-efficient nodes at the edge of the network and hence being located closer to the user [2,3]. However, compared to the classical remote cloud data centers, edge clouds have limited caching capacity and computational ability, which can restrict them from supporting a large number of concurrent MAR services. Therefore, an efficient utilization of storage and computational resources is required for MAR applications in an EC-supported wireless network.

Multi-view streaming applications have recently attracted increased popularity especially in virtual reality (VR) and augmented reality (AR) fields. In these types of applications, each viewpoint is a sequence of frames captured by a given camera and is usually treated as a single view stream. Through transmitting different high-quality view streams to the VR/AR equipment, the user could enjoy a more realistic perspective of 3D scenes [4,5]. However, multi-view applications are still confronted with limited resources in the network (e.g., the terminal, the wireless channel and cloud resources), and hence, it is not recommended to store

and transmit all view streams at a very high quality [6]. Thus, it is more efficient in terms of network resource utilization to endure a subset of view streams or a lower quality level during the service optimization process [4,7]. In this paper, a fixed given quality level is applied for the different view streams during modeling, and only popular ones are proactively cached and transmitted for multi-view streaming AR applications. To illustrate the use case, a toy example of the proposed work flow of multi-view streaming AR applications is detailed by Figure 1. When receiving a set of frames with recognized AROs, the target server searches the matched view stream embedded with the requested ARO in its local cache. Clearly, a series of view streams might contain the same requested ARO but only a popular subset (or in some instances, depending on available resources, the most popular one) is proactively cached and returned to the user according to the given network constraints.

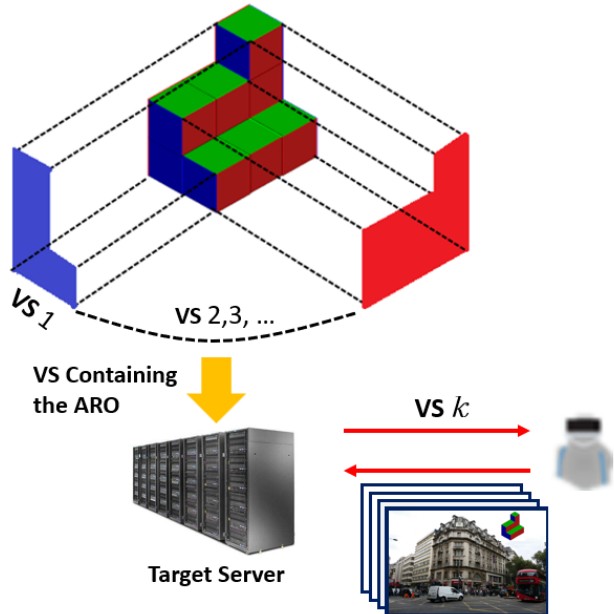

**Figure 1.** An example of multi-view streaming AR application.

Essentially, multi-view streaming MAR applications can be decomposed into two main types of functions that can operate in tandem (i.e., as a chain), known as storage intensive functions and computing intensive functions [8]. Service decomposition is also suitable for such applications because they are susceptible to storage and computing resources, especially when utilized in a high mobility scenario [8]. Figure 2 presents toy examples of whether considering service decomposition and multi-view streaming for MAR. In case (a), mobility is not considered, leading to a sub-optimal EC allocation. However, case (b) depicts the proposed mobility-aware allocation that considers also popular view streams and AROs. As shown in Figure 2, caching all AROs that are embedded in a multi-view streaming AR application at a single EC close to the user's initial location might be suitable only when there is no mobility event and there are sufficient EC resources. However, when taking mobility and congestion episodes, the different view streams and the number of AROs into account, it is more flexible and efficient to apply service decomposition and cache popular view streams and AROs into different ECs. Clearly, service decomposition allows a flexible assignment that fits better in a congested network with limited cache and computing resources. As shown, for example, by view stream 4 in case (b), if the target EC has limited resources, then for the selected view stream, it is recommended to cache only the most popular AROs. As a result, users request only popular view streams, such as for example view stream 4, as shown in the figure. It can be accessed with very low latency, since it can be cached in an EC at the user proximity. Hence, by exploring the popularity of the view streams and the AROs, a more efficient allocation can be implemented without affecting the overall user experience, since caching is taking

place by considering the most popular service composition for the end users. Hereafter, the view streams under consideration are assumed to be non real time, so that they could be cached at suitable EC locations during the service period [9]. It is also worth pointing out that in multi-view streaming AR applications, it is possible to identify popular view streams and associated AROs through historical data [8,10]. In that case, view streams and associated AROs per stream could be optimally anchored and decomposed in different EC by exploring, at the same time, the mobility pattern of the end users. In [11], a federated learning-based scheme is proposed to predict view ports based on the view history, and then, a stochastic game is formulated to optimize virtual reality video streaming through caching high or low-quality tiles of video according to given conditions. Similarly, considering latency and user preference, we aim to optimize the allocation of decomposed functions with popular view streams and embedded AROs on different target cloud servers.

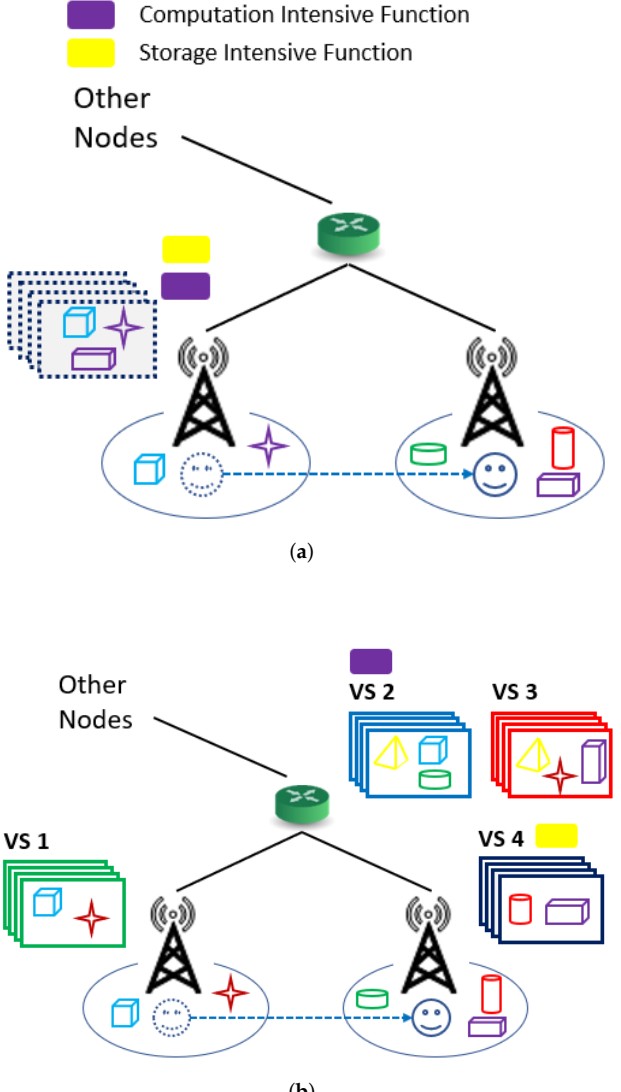

**Figure 2.** Illustrative toy examples where in case (**a**) mobility is not considered and (**b**) where mobility of the end user is taken into account for service decomposition and the different multi view streams on proactive resource allocation of MAR functions.

A Mixed Integer Linear Problem (MILP) is formulated and solved to obtain optimal solutions in this paper. However, due to the curse of dimensionality, it is not ideal for a large network with dynamic changes. Machine learning (ML)-based techniques can be deemed as an effective method for AR applications as they are able to learn from given data and provide

reliable predictions [12,13]. A well-known ML technique, the long short-term memory (LSTM) neural network, is utilized in this paper to learn from optimal solutions. The LSTM network resists gradient vanishing and gradient exploding by keeping the important information and key features in the long term while easily forgetting the unnecessary inferences [14,15]. In [16], LSTM is combined with a convolutional neural network to construct a supervised ML framework that improves the estimation of channel state information in 5G wireless systems. In this paper, we consider a typical Rayleigh fading channel to model the wireless link of a 5G access point, and the potential of multi-view streaming and service decomposition is explored through a nominal LSTM network.

In this paper, multiple view streams with embedded AROs are considered together with user mobility and MAR service decomposition to construct a joint optimization problem. The proposed framework seeks a balance between service latency and user preference of view streams and embedded AROs. Due to the combinatorial nature of the problem, the proposed optimization problem cannot be deemed as suitable for real-time decision making. Thus, we further propose an LSTM-based scheme to learn from optimal decision making in an offline manner to provide high-quality predictions during inference in an efficient manner suitable for real-time operation.

## 2. Related Work

This section presents the literature review which is closely relevant, whilst at the same time, we identify how the proposed algorithms approach and differ from previously proposed solutions.

In [17], the caching resource sharing problem is considered separately in a Radio Access Network (RAN) and the Core Network (CN) under the view of fully 5G virtualization. Similar to our work, a probability of requesting a set of popular content is assumed, and this information is used to formulate the actual cache hit ratio. More specifically, in [17], a dynamic Cournot game in RAN is firstly solved, and this provides a probability sorting algorithm to achieve the local optima for the CN. Then, a global equilibrium algorithm is proposed to achieve the balance between the given local optimal solutions [17]. Although the demand and competition in mobile virtual network operators are carefully researched in [17], the authors in [18] point out that the chaining of virtual network functions is also a key feature of 5G and beyond virtual networks and should be considered. In [18], a set of network slices is applied to generate a centralized optimization problem that maximizes system utility. A trade-off between efficiency and fairness could be achieved through selecting the utility function for their optimization problem [18]. A trade-off between latency and analytic accuracy of uploaded frames in an edge cloud-based MAR system is provided in [3]. In [19], another trade-off targeting on workload and service latency is provided. In their later work, two trade-offs are designed for the MAR system, where instead of choosing different target functions under different conditions, a weighting parameter is brought in to directly achieve the balance between latency and preference. Such an idea is also utilized in this paper. However, it has to be noted that most of the previously proposed schemes for MAR do not explicitly take into account wireless transmission, user mobility and service decomposition jointly.

There are other relevant works proposing improved frameworks and networking with the edge cloud support. In [20], a tensor-based cloud-edge computing framework is designed for cyber–physical–social systems, which can be separated to the cloud and edge plane. In their framework, the cloud manages large-scale, long-term and global data, while the edge servers react to real-time situations by processing small-scale, short-term and local data [20]. The proposed framework in this paper also tends to allocate decomposed services at edge servers while it turns to the further cloud when exceeding the caching or computing boundary. Another optimization framework in [21], alternating direction method of multipliers (ADMM), considers a general task and regards terminals, edge nodes and remote clouds as a three-tier network. Similar to our framework, this one also jointly optimizes offloading decisions and computing resource allocation to minimize the task

delay [21]. It then reduces problem complexity by decomposing the large-scale problem into smaller sub-problems and finally provides near-optimal solutions subjecting to the battery capacity [21]. The deep learning framework in [22] is specialized for AR applications and considers trade-offs between accuracy and latency. However, they accept a more powerful local "helper" (e.g., home servers), and hence, the AR application is processed either locally or in a remote cloud. MAR with Viable Energy and Latency (MARVEL) also accepts a simplified network structure including mobile AR devices and clouds but integrates local inertial tracking, local optical flow and visual tracking into a complete system [23].

At the application layer, there are relevant research works focusing on resource allocation problems for different video view streams. In [24], multiple views of a live video in a cloud-based system are considered in a similar fashion with this work. Similarly, they also try to optimize the overall users' satisfaction by formulating a Mixed Integer Linear Problem (MILP) with computational and bandwidth constraints. In addition, the paper proposes a Fairness-Based Representation Selection (FBRS) heuristic algorithm to achieve near-optimal results [24]. However, the focus in that work is mostly on the application layer quality of experience and received utility without emphasis on the wireless transmission process [24]. In [4], an interactive multi-view HTTP adaptive streaming is proposed with the emphasis in exploring the optimal subset of views under channel constraints. Due to the NP-hardness of the corresponding optimization problem, the paper offers the design of a sub-optimal low-complexity greedy algorithm with focus on network congestion episodes in hot spot areas. The work in [25] anchors its contribution in an optimization framework for the allocation of power and bandwidth in multi-user video streaming to improve the quality of user experience (QoE). Their work tracks requests in the unmanned aerial vehicle (UAV) relay networks and proposes a decentralized allocation scheme to derive the optimal solution [25]. The work in [26] shares a similar target with the current work. More specifically, in order to solve the pro-caching problem for vehicular multi-view 3D video streaming, the authors explicitly consider the mobility and construct a user mobility model that includes predicted moving direction and destinations. The difference is that they treat it as a Markov decision problem, and such a joint optimization problem concentrates on the overall reward and cost in 5G small-cell networks [26]. Due to the complexity of finding optimal decisions, a dynamic K-Nearest Neighbor algorithm is embedded into their proposed deep learning approach to provide the allocation for cache memory and the selection for the views set [26]. In [27], an object detection model of AR applications in an EC-supported network is considered, and the authors provide effective task placement and offloading strategies achieving the balance between latency and energy. Furthermore, the work in [27] directly identifies three latency classes and achieves the task execution time through historical data, while in this paper, a more detailed communication process is focused and calculated from the given formula.

## 3. System Model

### 3.1. Multi-View Streaming AR

We define with the set $\mathbb{M} = \{1, 2, \ldots, M\}$ the available edge clouds (ECs) in the wireless network. With $r \in \mathbb{R}$, we denote the MAR service requests that are generated by mobile users that are equipped with MAR devices (each user makes a single request). As already mentioned in previous sections, a set of AROs is assumed to be embedded across the different non real-time view streams. To this end, we first define a set $\mathbb{N} = \{1, 2, \ldots, N\}$ to represent the set of available AROs. The video content available to each user has multiple streams, and we define them as a set $\mathbb{S}_r = \{1, 2, \ldots, S\}$. Then, the decision variable for proactively caching a view stream $s$ at the EC $j \in \mathbb{M}$ is denoted as $p_{sj}$. The subset $\mathbb{L}_{rs}$ represents the target AROs required by the user $r$ in view stream $s \in \mathbb{S}_r$ and the size of each target ARO $l \in \mathbb{L}_{rs}$ is denoted as $O_l$. Finally, the decision variable for proactively caching an ARO required by a request $r$ is denoted as $h^s_{rl}$. More specifically, $p_{sj}$ and $h^s_{rl}$ can be written as follows,

$$p_{sj} = \begin{cases} 1, & \text{if view stream } s \text{ is cached at node } j, \\ 0, & \text{otherwise.} \end{cases} \tag{1}$$

$$h_{rl}^s = \begin{cases} 1, \text{ if ARO } l \text{ required by request } r \text{ embedded} \\ \quad \text{ in view stream } s \text{ is cached,} \\ 0, \text{ otherwise.} \end{cases} \tag{2}$$

in addition to the above, the following constraints should also be satisfied,

$$\sum_{r \in \mathbf{R}} h_{rl}^s \leqslant 1, \ \forall j \in \mathbf{M}, \ \forall s \in \mathbf{S_r}, \ \forall l \in \mathbf{L_{rs}} \tag{3}$$

$$\sum_{s \in \mathbf{S_r}} \sum_{l \in \mathbf{L_{rs}}} h_{rl}^s \geqslant 1, \ \forall r \in \mathbf{R} \tag{4}$$

$$\sum_{j \in \mathbf{M}} p_{sj} \geqslant h_{rl}^s, \ \forall r \in \mathbf{R}, \ \forall s \in \mathbf{S_r}, \ \forall l \in \mathbf{L_{rs}} \tag{5}$$

$$h_{rl}^s \leqslant h_{rl}^s \sum_{j \in \mathbf{M}} p_{sj}, \ \forall r \in \mathbf{R}, \ \forall s \in \mathbf{S_r}, \ \forall l \in \mathbf{L_{rs}} \tag{6}$$

The constraints in (3) ensure that each ARO can be cached at most once in a view stream. The constraints in (4) ensure that at least one view stream and an ARO is required to compose a valid request. The constraints in (5) guarantee that the ARO must be allocated to a view stream first before deciding to proactively cache it, and the constraints in (6) certify that any ARO planned to be stored in this view stream should not be cached when deciding not to proactively cache a view stream at all. Constraints (5) and (6) together ensure that either the view stream or ARO cannot be handled alone during the formulation.

As alluded earlier, due to the resource constraints in the network, in case of congestion episodes, an optimized subset of view streams and embedded AROs need to be proactively cached at suitable ECs. Thus, we bring in the user preference for both view streams and AROs. The preference of view streams and embedded AROs are measured by their requesting probability, which enables the proposed scheme to select the more popular ones. The probability of selecting a view stream $s$ is denoted as $t_s$ and the probability of requiring an ARO $l$ is denoted as $t_l$. Such probabilities are pre-defined and could be achieved by exploring historical data; hereafter we assume that we have access to such information. Thus, we further define $\mathbb{T}_\mathbb{l}$ and $\mathbb{T}_\mathbb{s}$, respectively to represent two descending probability vectors of AROs and view streams. The probability $t_l$ and $t_s$ inside can be generated through,

$$t_l = \begin{cases} 1, \text{ if } l = 1, \\ \text{rand } (t_{l-1}), \text{ if } l > 1. \end{cases} \tag{7}$$

$$t_s = \begin{cases} 1, \text{ if } s = 1, \\ \text{rand } (t_{s-1}), \text{ if } s > 1. \end{cases} \tag{8}$$

where the function rand($x$) generates a random number within $[0, x]$. Then, each element should be normalized through,

$$\frac{t_l}{\sum_{l \in \mathbf{L_{rs}}} t_l}, \ \frac{t_s}{\sum_{s \in \mathbf{S_r}} t_s} \tag{9}$$

### 3.2. Mathematical Programming Formulation

In a similar manner with [8], computational-intensive and storage-intensive MAR functionalities are defined as $\eta$ and $\varrho$, respectively. In addition, the corresponding execution location for a functionality is denoted as $x_{ri}$ and $y_{ri}$, respectively [8]. The starting location of the request $r$ is the access router where this user is initially connected to; this initial location is defined as $f(r)$. A user moves to a destination $k \in \mathbb{K}$ in the case of a mobility event (i.e., changing the point of attachment). In this paper, and without loss of generality, only adjacent access routers can be regarded as allowable destinations in the mobility event. Hence, $u_{f(r)k} \in [0, 1]$ is defined to represent the probability of a user moving from

the initial location to an allowable destination, where adjacent servers $\{f(r), k\} \subset \mathbb{M}$. Such probabilities can also be learned from historical data, which are readily available from mobile operators. The size of uploaded frames with target AROs used in the Object Detection function is denoted as $F_{\eta r}$ and the size of pointers used for identifying target AROs is denoted as $F_{\varrho r}$ [8,10]. During the matching process, the target AROs are possibly not pre-cached in the local cache and such case is known as a "cache miss" (otherwise there is a "cache hit"). Whenever confronted with a cache miss, the request is redirected to a core cloud deeper in the network and suffers from an extra latency $D$ as penalty.

At first, a joint optimization scheme considering the balance between the service delay and the preference of view streams and embedded AROs is designed to track the MAR requests in the EC-supported network. The cache hit/miss is captured by the decision variable $z_{rj}$ and can be written as follows,

$$
z_{rj} = \begin{cases} 1, \text{ if } \sum_{l \in \mathbf{L_{rs}}} \sum_{s \in \mathbf{S_r}} p_{sj} h_{rl}^s \geqslant L_{rs}, \\ 0, \text{ otherwise.} \end{cases} \tag{10}
$$

In [8], constraints (10b) and (10d) reveal the cache limitation and the cache hit/miss relation. In this paper, they should be rewritten as follows,

$$
\sum_{r \in \mathbf{R}} \sum_{l \in \mathbf{L_{rs}}} \sum_{s \in \mathbf{S_r}} p_{sj} h_{rl}^s O_l \leq \Theta_j, \forall j \in \mathbf{M} \tag{11}
$$

$$
\sum_{l \in \mathbf{N}} \sum_{s \in \mathbf{S_r}} h_{rl}^s + \epsilon \leq L_{rs} + U(1 - q_{rj}) \, \forall j \in \mathbf{M}, r \in \mathbf{R} \tag{12}
$$

where $\Theta_j$ is the cache capacity at node $j$. In (12), to rewrite the either-or constraint that $\sum_{l \in \mathbf{N}} \sum_{s \in \mathbf{S_r}} h_{rl}^s < L_{rs}$ or $z_{rj} = 1$, we define $\epsilon$ as a small tolerance value, $U$ as a large arbitrary number and $q_{rj}$ as a new decision variable satisfying $1 - q_{rj} = z_{rj}$ [8]. Requiring more pre-cached view streams and embedded AROs in a request naturally lead to an extra burden for the matching function. More specifically, the processing delay of the matching function can be written as,

$$
W_{rj} = \frac{\omega(F_{\varrho r} + \sum_{l \in \mathbf{L_{rs}}} \sum_{s \in \mathbf{S_r}} p_{sj} h_{rl}^s O_l)}{f_V^j} \tag{13}
$$

where $\omega$ is the computation load, $f_V^j$ is the virtual CPU frequency and $F_{\varrho r}$ are the size of the uploaded pointers [8,10]. When finding the target ARO during matching, other view streams containing it should also be transferred to the user. Hence, the delay of sending results back to the user can be written as follows,

$$
\sum_{s \in \mathbf{S_r}} \sum_{i \in \mathbf{M}} \sum_{j \in \mathbf{M}} C_{ij} y_{ri} p_{sj} + \sum_{s \in \mathbf{S_r}} \sum_{r \in \mathbf{R}} \sum_{j \in \mathbf{M}} \sum_{k \in \mathbf{K}} C_{jk} u_{f(r)k} p_{sj} \tag{14}
$$

where $u_{f(r)k}$ is the moving probability from the initial location $f(r)$ to a potential destination $k$. In previous expressions, the product of decision variables $p_{sj} h_{rl}^s$ and $p_{sj} y_{rj}$ exists and creates a non-linearity. In addition, note that when executing the matching function at the location $j$ ($W_{rj} y_{rj}$), the product of decision variables $p_{sj} h_{rl}^s y_{rj}$ also appears. To linearize the expressions above, so that to utilize linear integer programming solution methodologies, new auxiliary decision variables are brought in. A new decision variable $\alpha_{rsj}$ is introduced as $\alpha_{rsj} = p_{sj} y_{rj}$ and the constraints should be added as follows,

$$
\begin{aligned}
\alpha_{rsj} &\leqslant p_{sj}, \\
\alpha_{rsj} &\leqslant y_{rj}, \\
\alpha_{rsj} &\geqslant p_{sj} + y_{rj} - 1
\end{aligned} \tag{15}
$$

similarly, a new decision variable $\beta_{rslj}$ is introduced as $\beta_{rslj} = p_{sj}h_{rl}^s$ and the constraints should be added as follows,

$$
\begin{aligned}
\beta_{rslj} &\leqslant p_{sj}, \\
\beta_{rslj} &\leqslant h_{rl}^s, \\
\beta_{rslj} &\geqslant p_{sj} + h_{rl}^s - 1
\end{aligned}
\tag{16}
$$

the aforementioned constraint (6) is affected and should be rewritten as follows,

$$
h_{rl}^s \leqslant \sum_{j \in \mathbf{M}} \beta_{rslj}, \ \forall r \in \mathbf{R}, \ \forall s \in \mathbf{S_r}, \ \forall l \in \mathbf{L_{rs}}
\tag{17}
$$

In addition, note that $p_{sj}$ is a binary decision variable and causes $p_{sj} = p_{sj}^2$. Thus, we have $p_{sj}h_{rl}^s y_{rj} = \alpha_{rsj}\beta_{rslj}$. A new decision variable $\lambda_{rslj}$ is introduced as $\lambda_{rslj} = \alpha_{rsj}\beta_{rslj}$ and the following constraints should be added as follows,

$$
\begin{aligned}
\lambda_{rslj} &\leqslant \alpha_{rsj}, \\
\lambda_{rslj} &\leqslant \beta_{rslj}, \\
\lambda_{rslj} &\geqslant \alpha_{rsj} + \beta_{rslj} - 1
\end{aligned}
\tag{18}
$$

### 3.2.1. Wireless Channel Modeling and Achievable Data Rate

According to Shannon formula, the achievable wireless data rate for the user in a 5G access point is $B_j \log_2(1 + \gamma_{rj})$, where $B_j$ is the bandwidth allocated to the user's resource block and $\gamma_{rj}$ is the Signal to Interference plus Noise Ratio (SINR) of the user $r$ at the node $j$. We denote $P_j$ as the current resource block power, $H_{rj}$ as the channel gain, $N_j$ as the noise, $a$ as the path loss exponent and $d_{rj}$ as the Euclidean Distance between the user and the access router in the cell. Furthermore, we utilize a nominal Rayleigh fading channel to model the channel between a 5G access point and the users [28]. In this case, the channel gain $H_{rj}$ can be written as follows [29],

$$
H_{rj} = \sqrt{\frac{1}{2}(t + tJ)}
\tag{19}
$$

where $J^2 = -1$, $t$ is a random number following the standard normal distribution. Then, the SINR $\gamma_{rj}$ in a 5G wireless network can be written as follows [29,30],

$$
\gamma_{rj} = \frac{P_j H_{rj}^2 d_{rj}^{-a}}{N_j + \sum_{i \in \mathbf{M}, i \neq j} P_i H_{ri}^2 d_{ri}^{-a}}
\tag{20}
$$

Thus, the wireless transmission delay in a mobility event can be written as follows,

$$
\begin{aligned}
&\sum_{r \in \mathbf{R}} \frac{F_{\eta}r + \sum_{s \in \mathbf{S_r}} \sum_{l \in \mathbf{L_{rs}}} h_{rl}^s O_l}{B_{f(r)} \log_2(1 + \gamma_{rf(r)})} + \\
&\sum_{r \in \mathbf{R}} \sum_{k \in \mathbf{K}} u_{f(r)k} \frac{F_{\eta}r + \sum_{s \in \mathbf{S_r}} \sum_{l \in \mathbf{L_{rs}}} h_{rl}^s O_l}{B_k \log_2(1 + \gamma_{rk})}
\end{aligned}
\tag{21}
$$

### 3.2.2. Latency and Preference

Based on the above derivations and in line with [8], the service latency can be written as follows,

$$L = (21) + \sum_{r \in \mathbf{R}} \sum_{i \in \mathbf{M}} (C_{f(r)i} + V_{ri}) x_{ri} +$$

$$\sum_{r \in \mathbf{R}} \sum_{i \in \mathbf{M}} \sum_{j \in \mathbf{M}} (\frac{\omega(F_{\varrho r} y_{rj} + \sum_{l \in \mathbf{L_{rs}}} \sum_{s \in \mathbf{S_r}} \lambda_{rslj} O_l)}{f_V^j} +$$

$$C_{ij} \xi_{rij} + C_{jf(r)} y_{rj} + \psi_{rj} D) + \tag{22}$$

$$\sum_{s \in \mathbf{S_r}} \sum_{i \in \mathbf{M}} \sum_{j \in \mathbf{M}} C_{ij} \alpha_{rsj} + \sum_{s \in \mathbf{S_r}} \sum_{r \in \mathbf{R}} \sum_{j \in \mathbf{M}} \sum_{k \in \mathbf{K}} C_{jk} u_{f(r)k} p_{sj}$$

$L_{max}$ here denotes the maximum allowed service latency. Thus, we have,

$$\frac{L}{L_{max}} \in [0,1] \tag{23}$$

The overall preference of view streams and embedded AROs is measured by the summation of probabilities of the chosen view streams and AROs. It is denoted as $Q$ and can be written as follows,

$$Q = \sum_{s \in \mathbf{S_r}} \sum_{j \in \mathbf{M}} p_{sj} t_s + \sum_{r \in \mathbf{R}} \sum_{s \in \mathbf{S_r}} \sum_{j \in \mathbf{M}} \sum_{l \in \mathbf{L_{rs}}} p_{sj} h_{rl}^s t_l$$

$$= \sum_{s \in \mathbf{S_r}} \sum_{j \in \mathbf{M}} p_{sj} t_s + \sum_{r \in \mathbf{R}} \sum_{s \in \mathbf{S_r}} \sum_{j \in \mathbf{M}} \sum_{l \in \mathbf{L_{rs}}} \beta_{rslj} t_l \tag{24}$$

similarly, $Q_{max}$ here denotes the maximum allowable preference of view streams and embedded AROs. Hence, for the preference of view streams and embedded AROs, we also have,

$$\frac{Q}{Q_{max}} \in [0,1] \tag{25}$$

The weight parameter is denoted as $\mu \in [0,1]$, and the joint optimization problem can eventually be written as follows,

$$min \; \mu \frac{L}{L_{max}} - (1-\mu) \frac{Q}{Q_{max}} \tag{26a}$$

$$s.t. \; z_{rj} = 1 - q_{rj}, \; \forall j \in \mathbf{M}, r \in \mathbf{R} \tag{26b}$$

$$\sum_{r \in \mathbf{R}} (x_{rj} + y_{rj}) \le \Delta_j, \forall j \in \mathbf{M} \tag{26c}$$

$$\sum_{j \in \mathbf{M}} x_{rj} = 1, \forall r \in \mathbf{R} \tag{26d}$$

$$\sum_{j \in \mathbf{M}} y_{rj} = 1, \forall r \in \mathbf{R} \tag{26e}$$

$$\xi_{rij} \le x_{ri}, \; \forall r \in \mathbf{R}, i, j \in \mathbf{M} \tag{26f}$$

$$\xi_{rij} \le y_{rj}, \; \forall r \in \mathbf{R}, i, j \in \mathbf{M} \tag{26g}$$

$$\xi_{rij} \ge x_{ri} + y_{rj} - 1, \; \forall r \in \mathbf{R}, i, j \in \mathbf{M} \tag{26h}$$

$$\psi_{rj} \le z_{rj}, \; \forall r \in \mathbf{R}, j \in \mathbf{M} \tag{26i}$$

$$\psi_{rj} \le y_{rj}, \; \forall r \in \mathbf{R}, j \in \mathbf{M} \tag{26j}$$

$$\psi_{rj} \ge z_{rj} + y_{rj} - 1, \; \forall r \in \mathbf{R}, j \in \mathbf{M} \tag{26k}$$

$$x_{rj}, y_{rj}, p_{sj}, h_{rl}^s, z_{rj}, q_j \in \{0,1\},$$

$$\alpha_{rsj}, \beta_{rslj}, \lambda_{rslj}, \psi_{rj}, \xi_{rij} \in \{0,1\},$$

$$\forall r \in \mathbf{R}, j \in \mathbf{M}, l \in \mathbf{L_{rs}}, s \in \mathbf{S_r} \tag{26l}$$

$$(3), (4), (5), (11), (12), (15), (16), (17), (18)$$

As mentioned earlier, the constraint (26b) together with constraints (3) to (5), (11), (12) and (17) reveal the relation between pre-caching decisions and the cache miss/hit for each request [8]. The constraint (26c) is the virtual machine limitation while (26d) and(26e) guarantee the once execution of each function of a request at a single server as explained in [8]. The constraints (15), (16), (18) and (26f) to (26k) are auxiliary and required to solve the product of decision variables for linearization. Key variables defined in the formulation are shown with their description in the following Table 1.

**Table 1.** Notation.

| Parameter | Description |
|:---:|:---:|
| **N** | Set of all AROs |
| **M** | Set of edge clouds |
| **R** | Set of user requests |
| $\mathbf{S_r}$ | Set of view streams |
| **K** | Set of user destinations |
| $\mathbf{L_{rs}}$ | Set of target AROs for request $r$ in corresponding view stream $s$ |
| $\mathbf{T_l, T_s}$ | Accessing probability vectors of AROs and view streams |
| $\eta, \varrho$ | Two types of AR functions (CPU, Cache) |
| $O_l$ | Size of ARO $l$ |
| $h_{rl}^s$ | 0/1 var.: if ARO $l$ in view stream $s$ for the request $r$ is cached |
| $p_{sj}$ | 0/1 var.: if view stream $s$ is cached at node $j$ |
| $x_{rj}, y_{rj}$ | 0/1 var.: if function $\eta$ or $\varrho$ for $r$ is set at EC $j$ |
| $z_{rj}$ | 0/1 var.: if a cache hit is spotted for request $r$ |
| $C_{ij}$ | Communication delay between nodes $i$ and $j$ |
| $F_{\eta r}, F_{\varrho r}$ | Input size for functions $\eta, \varrho$ in request $r$ |
| $H_{rj}$ | Channel gain for request $r$ at node $j$ |
| $\gamma_{rj}$ | SINR for request $r$ at node $j$ |
| $B_j, P_j, N_j$ | Resource block bandwidth, power and noise at node $j$ |
| $\alpha$ | path loss exponent |
| $d_{rj}$ | Euclidean distance between user $r$ to node $j$ |
| $f(r)$ | Initial access point for request $r$ |
| $D$ | Cache miss penalty |
| $u_{f(r)k}$ | Moving probability from initial location $f(r)$ to destination $k$ |
| $\omega, f_V^j$ | Computation load and CPU availability at EC $j$ |
| $V_{rj}, W_{rj}$ | Processing delay for request $r$ for function $\eta, \varrho$ at EC $j$ |
| $L, Q$ | Overall latency and user preference |
| $\Theta_j$ | Cache capacity at EC $j$ |

*3.3. Heuristic Algorithm Using LSTM*

The aforementioned optimization problem could provide a series of optimal solutions for learning. These optimal solutions are grouped and reshaped into a route matrix and then separated for training and testing purposes. A nominal LSTM network is then designed to explore the potential of multi-view streaming and improve the computing efficiency. Its state changing and gate controlling also follow the nominal LSTM architecture. As shown in Figure 3, we feed user initial locations and destinations into the LSTM network as input and optimized locations of servers anchoring MAR functionalities as output. All possible different results can be calculated as selecting any two execution locations from the set $\mathbb{M}$ with order, which is $M^2$. Then, the LSTM-based model classifies and creates a mapping between the two sets. A Feasibility Check Stage is required afterwards to guarantee that predictions from the LSTM network could still satisfy the given constraints in the original network. If violating a constraint, such as exceeding an EC's available cache size for example, the remaining predicted requests served by this EC will be redirected to an available neighbor EC as a backup choice. However, if both ECs are fully occupied, such remaining requests have to be transmitted to a further core network cloud and hence suffer from an extra penalty in latency. Figure 4 shows a toy example of applying the LSTM network in our case. The LSTM network treats vectors with low appearance frequency such as [1,3]/[2,4] and

[3,1]/[2,2] as unimportant ones and forgets them during iterations. When several types of optimal solutions (e.g., [2,5] and [4,5]) for the same user route vector (e.g., [1,5]) share a similar level of appearance frequency, it is possible that the LSTM network generates a prediction between given solutions (e.g., [3,5]). The validation accuracy might still be acceptable, but in the Feasibility Check Stage, an activated neighbor EC with enough resources will substitute the current one (e.g., [2,5] replaces [3,5]).

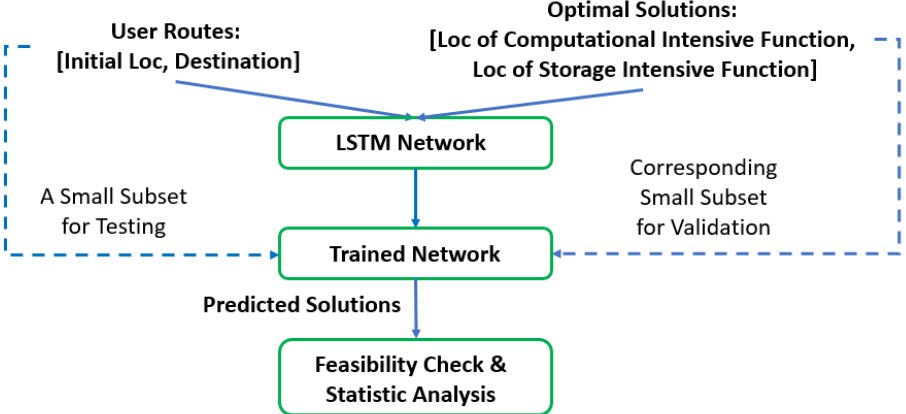

**Figure 3.** Working Process of the LSTM-based Model.

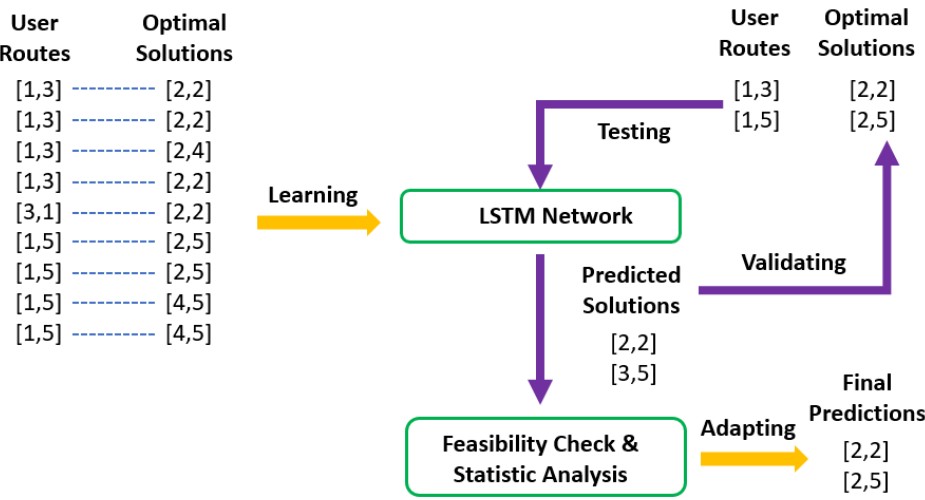

**Figure 4.** A Toy Example of the LSTM-based Model.

## 4. Numerical Investigations

Hereafter, the effectiveness of the proposed optimization scheme (noted as Optim) and the heuristic algorithm using the LSTM network (noted as LSTM) are investigated via a wide set of simulations and compared with a number of nominal/baseline schemes.

### 4.1. Parameterization

The same as in [8], a typical tree-like network topology as shown by Figure 5 is applied with 20 ECs in total with four to eight ECs being activated for the current MAR service, and 20 to 40 requests are sent by MAR devices. The remaining available resources allocated for MAR support of an EC are assumed to be CPUs with four to eight cores and 100 to 400 MBytes of cache memory [8]. Each request requires a single free unit for each service function (for example, a VM) [3]. There are 10 to 14 available VMs in each EC, with equal splitting of the available CPU resources [8]. Up to four different view streams of the video content per user can be cached and are similar to each other. All target AROs must be embedded within the corresponding view stream before being streamed to an end user

based on a matched result. The user could trigger the MAR service by a certain behavior (e.g., clicking an AR label), and pointers such as a name or index are transferred to the EC to identify the ARO [10]. The size of such pointers used for matching are only a few bytes in size, and hence, their transmission and processing latency is neglected in the simulations. The bandwidth of a nominal 5G access point is set to 20 MHz, its transmission power is assumed to be 20 dBm with the maximum of 100 resource blocks, and we assume, without loss of generality, each user can utilize only one resource block [31–33]. The noise power is $10^{-11}$ W and the path loss exponent is 4 [31]. The location of the users is randomly generated as well as the potential destinations. Furthermore, we  assume that each cell has a radius as 250 m in the 5G wireless network [31].  For the LSTM network, 90% of 300 route vectors with corresponding optimal decisions are utilized for training, while the remaining 10% are used for testing and validation. The initial learning rate is set to 0.005, the maximum number of epochs is 160 and the dropout probability is 5% to avoid over-fitting. Key simulation parameters are summarized in the following Table 2.

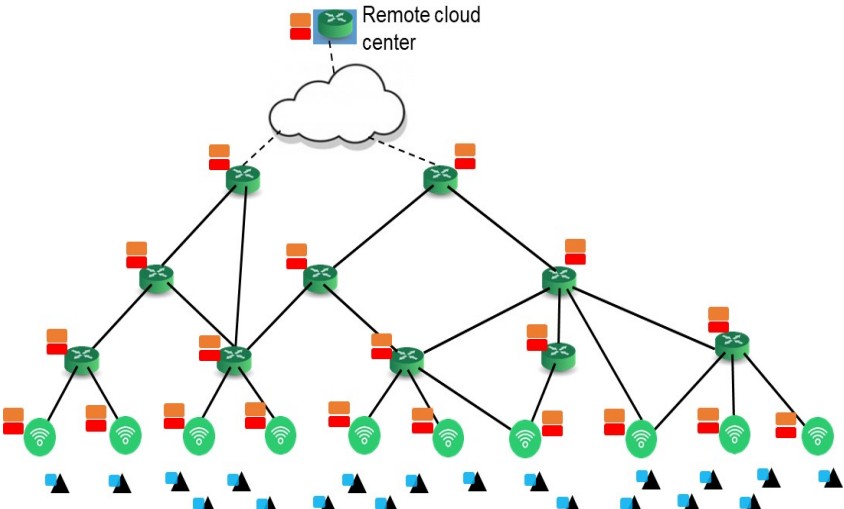

**Figure 5.** Typical tree-like designed network topology.

To appreciate the quality of the proposed set of solutions, a number of baseline schemes for caching MAR content are also included in the investigations. The Random Selection Scheme (RandS) allocates MAR content on randomly picked ECs. The Closest First Scheme (CFS) directly chooses the closest edge cloud to the user's initial location and accepts the second closest one as a backup choice [34]. The utilization based scheme (UTIL) selects the closest edge cloud as the CFS scheme but takes the least loaded EC as its backup choice and rejects any more cache or execution decisions if an EC exceeds its acceptable operational state (for example over 80% occupied VMs in the EC) [35].  Note that all such baseline schemes assign view streams and embedded AROs according to the above mentioned policies and hence neglect the user mobility (i.e., they are mobility oblivious). Furthermore, they will also suffer from a penalty cost if their target ECs (including backup EC choices) are all fully occupied.

**Table 2.** Simulation parameters.

| Parameter | Value |
| --- | --- |
| Number of available ECs | $[4, 8]$ |
| Number of available VMs per EC (EC Capacity) | $[10, 14]$ |
| Number of requests | $[20, 40]$ |
| Number of view streams per user | 4 |
| AR object size | $(0, 20]$ MByte |
| Total moving probability | $[0, 1]$ |
| Cell radius | 250m |
| Remained cache capacity per EC | $[100, 400]$ MByte |
| CPU frequency | $[2, 4]$ GHz |
| CPU cores | $[4, 8]$ |
| CPU portion per VM | 0.125–0.25 |
| Computation load | 10 cycles/bit |
| Bandwidth of access router | 20 MHz |
| Power of access router | 20 dBm |
| Path loss exponent | 4 |
| Noise power | $10^{-11}$ dBm |
| Number of resource blocks | 100 |
| Average hop's latency | 2 ms |
| Cache miss penalty | 25 ms |
| Number of user route vectors/optimal decisions | 300 |
| Initial LSTM learning rate | 0.005 |
| Maximum number of epochs | 160 |
| LSTM dropout probability | 5% |

*4.2. Simulation Results and Discussion*

According to Figure 6, an increasing weight ($\mu$) leads to a dropping overall delay as expected. A larger weight means that the proposed Optim scheme targets a solution with less latency and is willing to sacrifice some preference for this in terms of importance. As a result, the Optim scheme tends to proactively cache fewer view streams with fewer embedded AROs, and such a decision naturally results in a reduced overall delay in transmission and computation. As shown by Table 3, it is interesting to point out that when the emphasis is placed solely in delay reduction ($\mu = 1$) and without considering user mobility, the proposed Optim scheme closely relates to the other greedy baseline schemes. This means those schemes commonly tend to anchor view streams with embedded AROs as close as possible to the user. However, the Optim scheme is still superior to others and is around 15.5% better than the CFS scheme in this case because its service decomposition and separation of view streams enables flexible allocation and hence avoids overloading. In addition, it is worth pointing out that solutions offered from other greedy schemes sometimes provide non-feasible solutions due to constraints violations. Thus, the Optim scheme significantly outperforms other schemes especially in a high mobility scenario and/or in the cases where changes of the point of attachment take place during the service time. The observed gains between the proposed Optim scheme and other baseline schemes also increase with overall delay when continuously seeking a larger preference and caching a higher number of view streams and AROs. Because the UTIL scheme prevents further transmitted requests when the activated ECs reach the 80% occupation limit, it becomes the most sensitive scheme to the weight $\mu$ and hence its performance is heavily affected in a congested network.

Figure 7 shows the relation between preference and weight ($\mu$) and further reveals the advantage of the proposed Optim scheme in preference. Noticing that even in the extreme case when $\mu = 1$, the proposed Optim scheme still achieves approximately 59.6% preference due to its optimized selection of the most preferred view streams with the most popular AROs set inside. This means the proposed Optim scheme minimizes the cost of weight by exploring preference when seeking to reduce service latency.

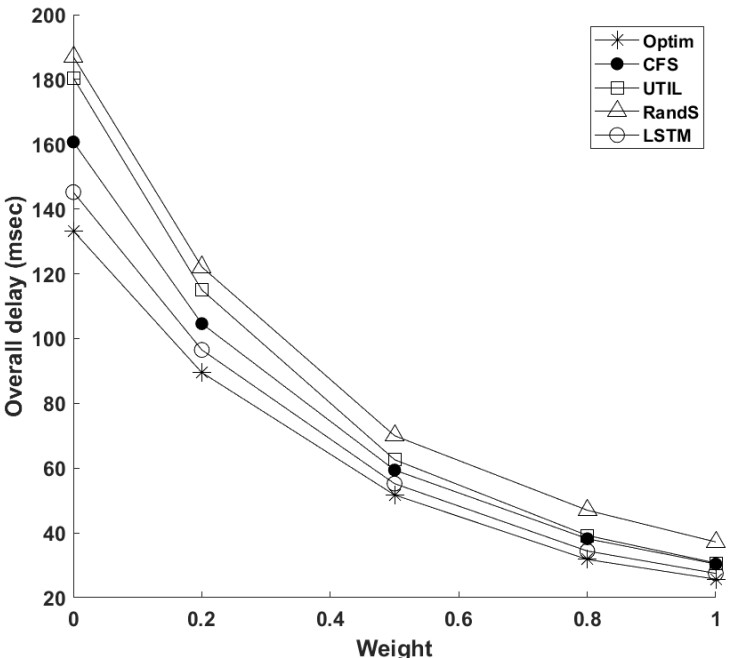

**Figure 6.** Service delay with weight $\mu$ (6 EC, 30 requests, EC capacity is 14 and total mobility probability is 1). CFS [34]; UTIL [35].

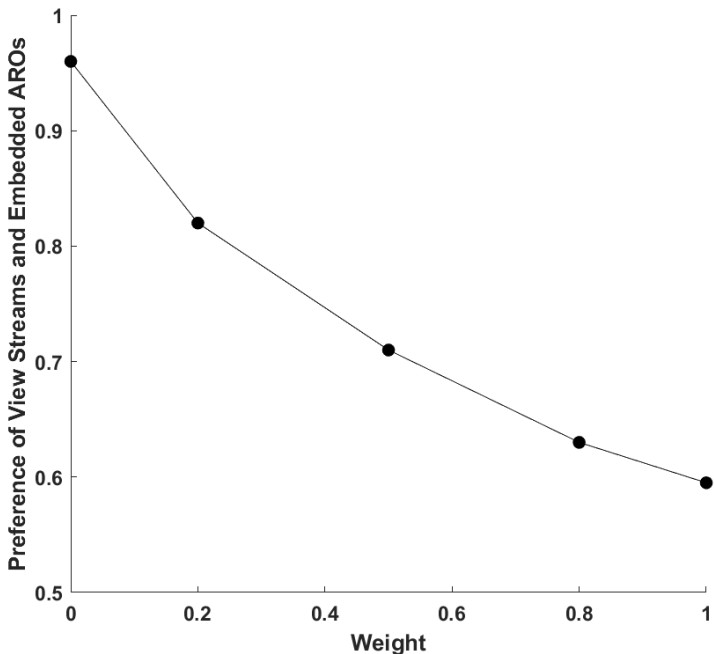

**Figure 7.** Preference of View Streams and Embedded AROs with Weight $\mu$ (6 EC, 30 Requests, EC Capacity is 14 and Total Moving Probability is 1).

According to Figure 8, the service delay increases when the network congestion levels increase due to a higher number of requests from the users. The LSTM scheme shows a 5.6%

to 9.8% optimality gap compared to the Optim scheme and approaches the Optim scheme better in a non-congested condition. The above aspect can also be seen from Figures 9 and 10. All schemes suffer from an increased delay when fewer ECs are activated or there is less available capacity per EC. However, in all use cases under investigation, the proposed schemes show much better performance in latency than other baseline schemes.

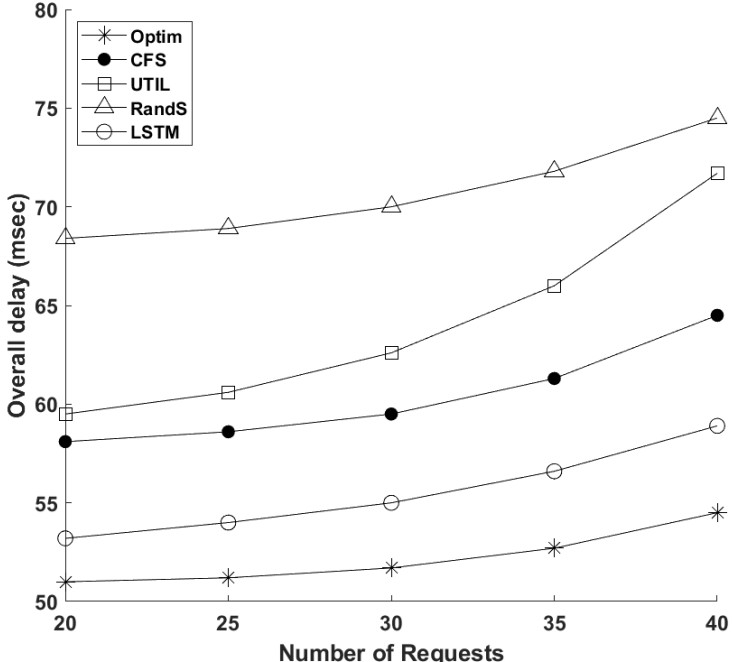

**Figure 8.** Service Delay with Different Number of Requests (6 EC, $\mu = 0.5$, EC Capacity is 14 and Total Moving Probability is 1). CFS [34]; UTIL [35].

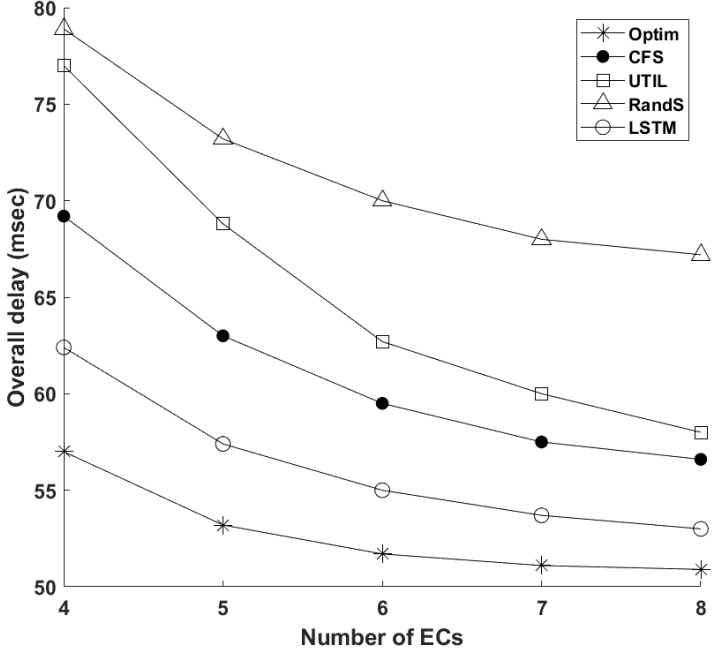

**Figure 9.** Service Delay with Different Number of ECs (30 Requests, $\mu = 0.5$, EC Capacity is 14 and Total Moving Probability is 1). CFS [34]; UTIL [35].

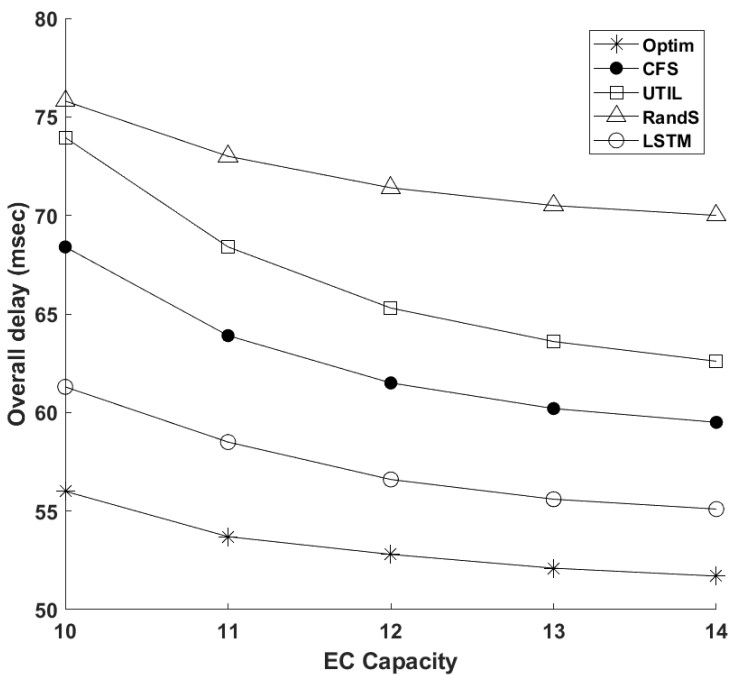

**Figure 10.** Service Delay with Different EC Capacities (6 EC, 30 Requests, $\mu = 0.5$ and Total Moving Probability is 1). CFS [34]; UTIL [35].

**Table 3.** RMSE values and service delay in no mobility event ($\mu = 1$, 6 ECs, 30 requests and EC capacity is 14).

| Scheme | Optim | LSTM | CFS | UTIL | RandS |
|---|---|---|---|---|---|
| **Delay (ms)** | 25.7 | 27.9 | 30.2 | 31.1 | 37.4 |
| **RMSE** | - | 2.4 | 8.5 | 3.6 | 5.8 |

To shed light on the performance, we calculate the *rmse* (root mean square error), $\delta$ (relative error) and $r^2$ (determination coefficient) for the LSTM scheme. The predictions from the LSTM network are compared with the optimal solutions, and we have $rmse < 2.5$, $\delta < 15\%$ (relative error) and $r^2 > 0.83$ (determination coefficient) in most cases. Hence, the predictions of the LSTM scheme can be regarded as reliable and closely follow the optimal solutions. Table 3 shows *rmse* values including the LSTM scheme and greedy baseline schemes for comparison. In simulations, topologically close servers are allocated adjacent indices, and the user mobility is limited to the adjacent server in a period of an MAR session. Hence, indices decided for each server also reveal their location information. Thus, the similarity between the predictions of the LSTM scheme and the optimal solutions reveals the LSTM scheme's competitive edge over other greedy baseline schemes. Noticing that an infeasible prediction might appear and cause a deteriorating quality of service after adjusted in the Feasibility Check Stage. However, according to Figure 11, the LSTM network's validation accuracy is maintained at around 86.3% and is less than 6% worse than its training accuracy, which indicates that the infeasible solutions are rare and the general learning capabilities are effective.

$$rmse = \sqrt{\frac{\Sigma_1^n(YTest_i - YPred_i)^2}{n}} \tag{27}$$

$$\delta = \frac{\Sigma_1^n |\frac{YTest_i}{YPred_i} - 1|}{n} \times 100\% \tag{28}$$

$$r^2 = 1 - \frac{(YTest_i - YPred_i)^2}{YPred_i - \frac{1}{n}\Sigma_{i=1}^n YPred_i} \tag{29}$$

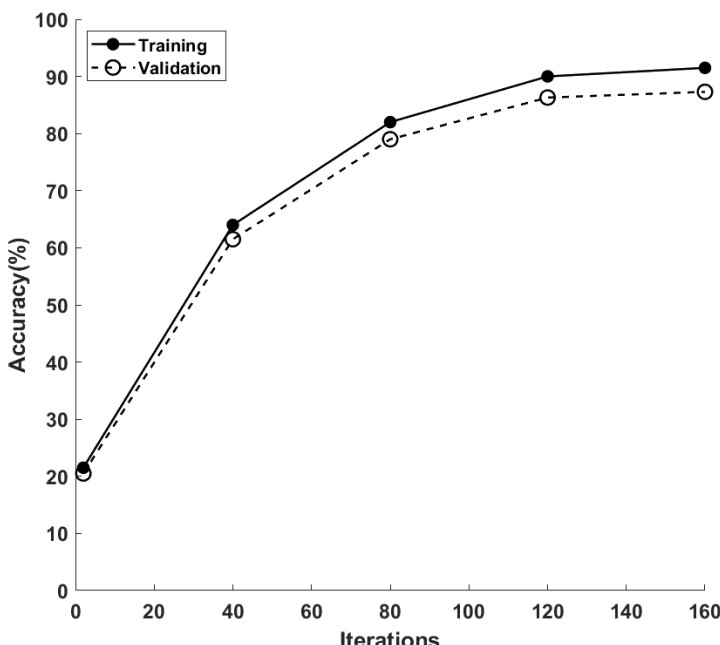

**Figure 11.** Average Training and Validation Accuracy of the LSTM Scheme through Iterations.

Before training the LSTM scheme, it requires a long computational time to create a large number of optimal solutions (~810 s in Figure 12) from the Optim scheme. However, this stage only needs to be executed a few times and is a purely offline task. Similarly, the network training (~70 s in Figure 12) is also offline and does not need to repeat itself if the validation accuracy of the trained LSTM network meets the expectation. As a result, these offline stages do not have a serious detrimental effect on the general computing efficiency. As shown in Table 4, based on a trained LSTM network, the online decision-making stage of the LSTM scheme can generate predictions in an efficient manner for MAR sessions. Because the Optim scheme constructs and solves a complex mixed integer linear problem, it suffers from the longest processing time and is not ideal for a highly dynamic network environment. The greedy baseline schemes own the obvious advantage in average processing time, but as discussed earlier, they cannot approach optimal solutions as well as the decision-making quality offered by the LSTM scheme.

**Table 4.** Average processing time of algorithms.

| Algorithm | Average Processing Time (s) | STD |
|---|---|---|
| RandS | 0.696 | 0.284 |
| CFS | 0.731 | 0.295 |
| UTIL | 0.753 | 0.308 |
| LSTM | 1.386 | 0.223 |
| Optim | 168.277 | 15.392 |

* tested by PC, intel i7, 1.3 GHz, 4 processors.

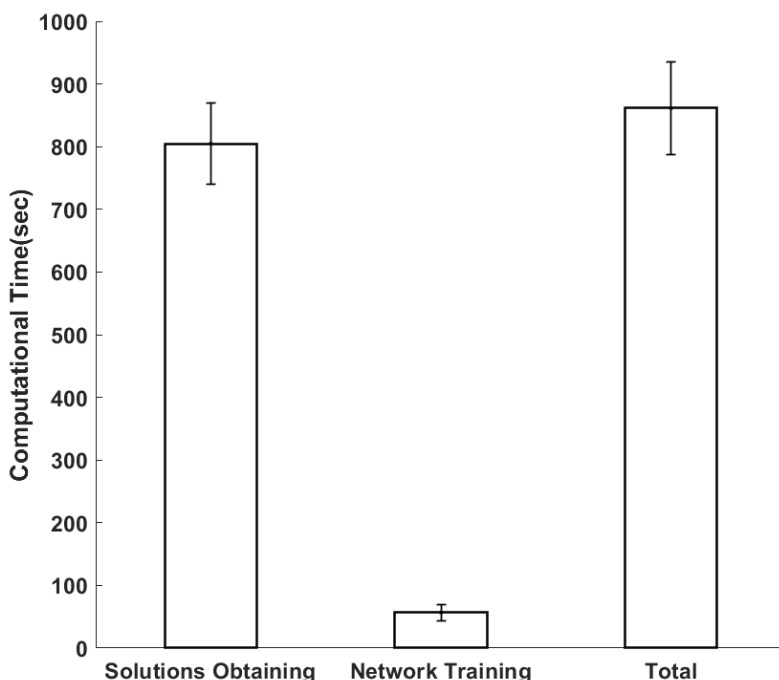

**Figure 12.** Average Processing Time for Different Stages in the LSTM Scheme.

## 5. Conclusions

Multi-view streaming applications with embedded augmented reality objects (AROs) are highly delay sensitive and confronted with a series of constraints in terms of storage and computing resources. In this paper, an optimization framework is proposed to realize optimal service decomposition in an edge-cloud supported 5G and beyond wireless network aiming to balance service latency and content popularity by taking into account the inherent user mobility. In addition, a nominal long short-term memory (LSTM) neural network is proposed that is trained using optimal solutions to explore real-time decision making for supporting edge proactive caching for multi-view streaming with popular augmented reality objects. It is important to note that the proposed framework is generic enough to be applied in a plethora of mobile-augmented reality applications that are composed by multiple view streams and popular augmented objects per stream. Numerical investigations reveal that the proposed schemes outperform a number of previously defined baseline schemes. More specifically, gains up to 38% are reported in terms of reducing the delay compared to previously proposed techniques that do not explicitly take into account user mobility and the inherent decomposition of the MAR services. in the proposed framework, during the online inference stage, the LSTM-based scheme is able to efficiently provide competitive predictions approaching the optimal solutions. On the other hand, the learning and training stage of the LSTM network require longer processing times, but those phases can be executed in an offline manner where high-quality solutions from the defined optimization framework could be explored. In that way, the proposed framework is able to provide high-quality decision making in an online manner by exploring the capabilities of the LSTM neural network and the powerful integer programming formulation that is able to provide optimal decision making that can be used for training the neural network.

**Author Contributions:** Z.H.; writing—original draft, writing—review and editing, V.F.; writing—review and editing, supervision. All authors have read and agreed to the published version of the manuscript.

**Funding:** This research received no external funding.

**Institutional Review Board Statement:** Not applicable.

**Informed Consent Statement:** Not applicable.

**Data Availability Statement:** All data included in this study are available upon request by contact with the corresponding author.

**Conflicts of Interest:** The authors declare no conflict of interest.

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
