# Peer review of "Optimal Proactive Caching for Multi-View Streaming Mobile Augmented Reality"

_futureinternet, doi:10.3390/fi14060166_

Round 1

Reviewer 1 Report

May 6, 2022

General Comments
In this paper, the Authors have analyzed the Mobile Augmented Reality (MAR) application to provide optimal service decomposition in an edge-cloud supported 5G and beyond wireless network aiming to balance service latency and content popularity.

Detailed Comments
I find the idea presented in the paper interesting and promising but I have the following comments and concerns:  
A. The Abstract should be rewritten more precisely. It should be pointed out/stressed measurably the effectiveness of the proposed approach. Moreover, the general conclusion which follows from the conducted analysis should be attached.

B. When reading this interesting and inspiring article, there seems to be a visible lack of a reference to the Free Space Optics Communication. Recently, an interesting analysis concerning these issues was developed in the articles

- Pang, X.; Ozolins, O.; Zhang, L.; Schatz, R.; Udalcovs, A.; Yu, X.; Jacobsen, G.; Popov, S.; Chen, J.; Lourdudoss, S, Free-Space Communications Enabled by Quantum Cascade Lasers. Phys. Status Solidi A 2021, 218: 2000407;
- Spitz, O.; Herdt, A.; Wu, J. et al. Private communication with quantum cascade laser photonic chaos. Nat. Commun. 2021, 12, 3327;
- Gaji´c, A.; Radovanovi´c, J.; Vukovi´c, N.; Milanovi´c, V.; Boiko, D.L. Theoretical approach to quantum cascade micro-laser broadband multimode emission in strong magnetic fields. Phys. Lett. A 2021, 387, 127007;
- Garlinska, M.; Pregowska, A.; Gutowska, I.; Osial, M.; Szczepanski, J. Experimental Study of the Free Space Optics Communication System Operating in the 8–12 μm Spectral Range. Electronic 2021, 10, 875;
- Lionis, A.; Peppas, K.; Nistazakis, H.E.; Tsigopoulos, A.D.; Cohn, K. Experimental Performance Analysis of an Optical Communication Channel over Maritime Environment. Electronics 2020, 9, 1109;
- Wang, Y.; Xu, H.; Li, D.; Wang, R.; Jin, C.; Yin, X.Y.; Gao, S.; Mu, Q.; Zuan, L.; Cao, Z. Performance analysis of an adaptive optics system for free-space optics communication through atmospheric turbulence. Sci. Rep. 2018, 8, 1124.
I recommend adding (including) into the paper a Paragraph, dealing with this issue among others on the above references.

C. The quantitative comparison with other existing literature algorithms (papers) should be added in Section: Discussion.

D. The language should be carefully revised. 

Final Comments
The idea presented and developed in the paper seems to be interesting and the results obtained are promising, but due to the above concerns, I would not recommend this paper for publication until the above comments/questions will be carefully addressed. At this moment I would recommend Major Revision.

Author Response

Please see the attachment (Reviewer 1).

Reviewer 2 Report

The proposed manuscript offers a Long Short Term Memory (LSTM) neural network to tackle the efficient amalgamation of Augmented Reality Objects(ARO) for Mobile Augmented Reality (MAR) applications.  

The proposed manuscript has clear merits and is an interesting read; however as a general remark I would suggest the authors improve the introduction and conclusion sections, and the correlation between these two. I didn’t actually understand consistently what the manuscript proposes: a new framework, a new network, or both.

Second general remark would concern the Related work section where more similar frameworks and networks should be considered. 

Third remark is on the Conclusion section which is too succinct. I advise the authors to expand this section of the manuscript.

Author Response

Please see the attachment (Reviewer 2).
